# Randomized Feasibility Pilot Trial of Adding a New Three-Dimensional Adjustable Posture-Corrective Orthotic to a Multi-Modal Program for the Treatment of Nonspecific Neck Pain

**DOI:** 10.3390/jcm11237028

**Published:** 2022-11-28

**Authors:** Ahmed S. A. Youssef, Ibrahim M. Moustafa, Ahmed M. El Melhat, Xiaolin Huang, Paul A. Oakley, Deed E. Harrison

**Affiliations:** 1Department of Rehabilitation Medicine, Tongji Hospital, Tongji Medical College, Huazhong University of Science and Technology, 1095#, Jiefang Avenue, Wuhan 430030, China; 2Basic Science Department, Faculty of Physical Therapy, Beni-Suef University, Beni-Suef 62521, Egypt; 3Department of Physiotherapy, College of Health Sciences, University of Sharjah, University City, Sharjah 27272, United Arab Emirates; 4Department of Physical Therapy for Musculoskeletal Disorders and their Surgeries, Faculty of Physical Therapy, Cairo University, Cairo 12613, Egypt; 5Department of Physical Therapy, Faculty of Health Sciences, Beirut Arab University, Beirut P.O. Box 11-5020, Lebanon; 6Independent Researcher, Newmarket, ON L3Y 8Y8, Canada; 7CBP NonProfit, Inc., Eagle, ID 83616, USA

**Keywords:** neck pain, orthotic, mirror image^®^ therapy, reverse posture training

## Abstract

The aim of this study was to investigate the feasibility and effect of a multimodal program for the management of chronic nonspecific neck pain CNSNP with the addition of a 3D adjustable posture corrective orthotic (PCO), with a focus on patient recruitment and retention. This report describes a prospective, randomized controlled pilot study with twenty-four participants with CNSNP and definite 3D postural deviations who were randomly assigned to control and study groups. Both groups received the same multimodal program; additionally, the study group received a 3D PCO to perform mirror image^®^ therapy for 20–30 min while the patient was walking on a treadmill 2–3 times per week for 10 weeks. Primary outcomes included feasibility, recruitment, adherence, safety, and sample size calculation. Secondary outcomes included neck pain intensity by numeric pain rating scale (NPRS), neck disability index (NDI), active cervical ROM, and 3D posture parameters of the head in relation to the thoracic region. Measures were assessed at baseline and after 10 weeks of intervention. Overall, 54 participants were screened for eligibility, and 24 (100%) were enrolled for study participation. Three participants (12.5%) were lost to reassessment before finishing 10 weeks of treatment. The between-group mean differences in change scores indicated greater improvements in the study group receiving the new PCO intervention. Using an effect size of 0.797, α > 0.05, β = 80% between-group improvements for NDI identified that 42 participants were required for a full-scale RCT. This pilot study demonstrated the feasibility of recruitment, compliance, and safety for the treatment of CNSNP using a 3D PCO to a multimodal program to positively affect CNSNP management.

## 1. Introduction

Chronic nonspecific neck pain (CNSNP) is a common musculoskeletal disorder worldwide. Because of CNSNP, disability-adjusted life years increased from 17 million (95% confidence interval (CI), 11.4–23.7) in 1990 to 29 million (95% CI, 19.5–40.5) in 2016 [1,2]. Treatment of CNSNP according to the clinical guidelines of APTA [3] includes manual therapy, therapeutic exercises, and posture education or correction. A study by Bernal-Utrera et al., 2020 concluded that manual therapy achieved a faster reduction in pain perception than therapeutic exercise, but therapeutic exercise reduced disability faster than manual therapy [4].

Although the exact relationship between posture and CNSNP is unsettled, poor posture of the cervical spine appears to influence dorsal neck muscle activity at rest and during movement [5]. Additionally, forward head posture (FHP) is associated with thoracic hyper-kyphosis and indirectly affects cervical flexion and the rotational range of motion (ROM) [6]. In addition, sustained computer work and prolonged use of smartphones appear to modify neck posture, as well as scapular positioning and upper trapezius muscle activity [7,8].

A recent systematic review conducted by Szczygieł et al., 2020 found that the posture of the head has a significant effect on the human body [9]. Abnormal head positions affect muscle activity, proprioception, and respiratory patterns and contribute to neck pain [9]. Another review, by Anabela et al., 2009 [10], concluded that head posture assessment is useful for neck patients, but it must be considered in relation to the patient’s symptoms and related functional problems [11,12,13].

Harrison [14,15] detailed posture displacements of the head, ribcage, and pelvis in three dimensions (3D) as translations and rotational displacements. Therefore, 3D postural assessment and correction during the treatment of CNSNP or postural neck pain should be considered [9,11]. There is also a growing body of research regarding patients with spinal dysfunction using mirror image^®^ therapy, which is prescribed specifically to help normalize the patient’s neuromuscular dysfunction and postural deformation by reflecting the patient’s posture across different planes [14,15,16,17]. The majority of interventions for improvement of abnormal posture focus on single or double combination movements (e.g., 1 or 2 movements at a time) as it is difficult to maneuver a patient’s head and neck in multiple planes and postures at once [14,15,16,17].

In the current investigation, we designed an adjustable 3D posture corrective orthotic (PCO) for the patient to wear for a short time (patent number CN201921929736.1). The PCO has the ability to reflect all translation and rotational displacements of the head in combination (3D planes). This mirror image therapy is designed to be delivered via the use of the adjustable PCO while the patient is walking at approximately 2–3 miles per hour on a standard, motorized treadmill. The PCO reverses the poor posture according to the 3D posture analysis data.

To the best of our knowledge, no randomized controlled trial (RCT) has evaluated the addition of a 3D adjustable PCO to a care program and investigated the short-term improvement effects on CNSNP management outcomes. The primary aim of our study was to perform a pilot RCT investigation to evaluate the feasibility of conducting a full-scale RCT considering recruitment, compliance to study protocols, adverse events, adherence, sample size calculation, and safety. The secondary aim was to investigate the effect size of adding the 3D PCO for mirror image therapy (reverse posture training) while the patient is walking on a motorized treadmill compared to a control group receiving standard interventions for neck pain intensity, disability, active ROM and 3D posture parameters after 10 weeks of intervention.

## 2. Methods

### 2.1. Study Design

This study was a double-blind (the different investigators and outcome assessor were blinded to group allocation) superiority pilot RCT with 2 parallel groups. The study was performed according to CONSORT guidelines. The ethics committee of Tongji Hospital, Tongji Medical College, Huazhong University of Science and Technology (HUST), Wuhan, China, approved the study protocol (certificate of approval number TJ-IRB20170703), which was prospectively registered at clinicaltrials.gov (Id: NCT03331120). The study was performed in the rehabilitation department at Tongji Hospital, affiliated with HUST, China.

### 2.2. Procedures

Participants were recruited through advertisements in orthopedic and rehabilitation department clinics and via mobile applications, such as WeChat (Tencent Ltd., Shenzhen, China). Participants first completed a written informed consent form, provided demographic data, and completed patient-reported outcome measures, including numeric pain rating scale score (NPRS) and neck disability index (NDI). Then, the outcome assessor measured the rotational and translational displacements of the head in relation to the thoracic region using a global postural system (GPS) device. After eligibility confirmation, another research assistant randomized participants to either a study or a control group receiving standard interventions only using sealed numbered envelopes using a randomization list generated by a random integer generator (www.random.org). A blinded investigator performed all outcome assessments at baseline and after 10 weeks of intervention. Participants were not blinded to their group allocation because of the difference in interventions between the two groups.

### 2.3. Participants

#### 2.3.1. Inclusion Criteria

Male and female participants aged 17–40; Ability to continue treatment for 10 weeks; Signature on informed consent form; Neck pain that was equal to or greater than 3/10 on the NPRS and pain lasting more than 3 months (chronic neck pain) [18,19]; A neck disability score on the NDI of at least 5 from a total score of 50 [20]; A 3D postural assessment, known as the Global Posture System (GPS) 600, (Chinesport, Udine, Italy).

#### 2.3.2. Posture Translations Displacements Included the Following

Anterior head translation (Tz) more than 2.5 cm [21,22]; Side shifting of the head (Tx) in relation to the thoracic region of more than 0.5 cm [16].

#### 2.3.3. Posture Rotations Displacements Included the Following

Rotation of head about vertical gravity (Ry) more than or equal to 3° [23]; Side bending of head (Rz) more than or equal to 3° [10]; Flexion or extension position of head (Rx). The average angle is 18°; if the angle was greater than this, it means extension in the upper cervical region, and if it is less, it means flexion in the upper cervical region [10]. Participants were included if they had at least two posture displacements, whether they were translations or rotations. We included more obvious amounts of translations of the head posture related to the thoracic region to avoid variability of measurement between participants such that the posture deviations could be visually examined. The mean absolute differences within examiners’ measurements (MADOMs) were 0.4 cm or less for lateral translations (Tx Head, Tx Thoracic, and Tx Pelvic) and 0.71 cm or less for forward translational measurements (Tz Head, Tz Thoracic, and Tz Pelvic). The MADOMs were 3.2° or less for flexion-extension rotational measurements (Rx Head, Rx Thoracic, and Rx Pelvic) and 1.4° or less for all axial rotations (Ry Head, Ry Thoracic, and Ry Pelvic) and lateral bending rotations (Rz Head, Rz Thoracic, and Rz Pelvic) [23].

#### 2.3.4. Exclusion Criteria

Neck pain associated with whiplash injuries, medical red flag history (such as tumor, fracture, metabolic diseases, rheumatoid arthritis, or osteoporosis) [19]; Neck pain with cervical radiculopathy or associated with externalized cervical disc herniation [19]; Fibromyalgia syndrome, because its diagnosis is similar to that of CNSNP [24]; Surgery in the neck area, regardless of the cause neck pain accompanied by vertigo caused by vertebra-basilar insufficiency or accompanied by non-cervicogenic headaches [19]; Current pain treatment, psychiatric disorders, or another problem that would contraindicate the use of the techniques in this study [19]; Any of the following conditions: (1) history of cervical or facial trauma or surgery, (2) congenital anomalies involving the spine (cervical, thoracic, or lumbar), (3) bony abnormalities, such as scoliosis, (4) any systemic arthritis, (5) recurrent middle ear infections over the last 5 years or any hearing impairment requiring the use of a hearing aid, (6) persistent respiratory difficulties over the last 5 years that necessitated absence from work, required long-term medication, or interfered with daily activities, (7) any visual impairment not corrected by glasses, (8) any disorder of the central nervous system, or (9) pregnancy or breast-feeding because these conditions affect head posture [25,26,27,28]; Inability to attend a 10-week treatment program.

### 2.4. Examination Procedures

All participants had histories taken. The history included demographic variables (age, sex, the mode of onset, duration of symptoms, nature and location of symptoms, and mechanism of injury, if it happened previously), as well as questions regarding aggravating and relieving factors, such as posture modifications and changed positions and any prior history of NP. All patients had a recent MRI study, no more than two weeks before the start of the study. In addition to the MRI. In addition to the MRI, other tests were performed to rule out the presence of space-occupying masses such as tumors, extruded intervertebral disks, osteophytes, nerve root irritation, or radiculopathy (specific neck pain), such as the Valsalva test, Spurling test (Foraminal compression test), distraction test, and Jackson compression test, followed by reflexes, cutaneous distribution, joint play movements, palpation, and diagnostic imaging. All participants underwent a physical examination. Pain level, neck function, and 3D posture analysis of the head in relation to the thoracic region and active cervical range of motion were measured before and after treatment [3,29].

#### 2.4.1. Primary Outcomes

The primary outcome of our study was to determine the feasibility of conducting an RCT; thus, we monitored the integrity of the study protocol, recruitment and retention, randomization procedures, primary outcome measures, and the sample-size calculation [30,31,32]. Further details and the results of each aspect of the primary outcomes are provided in the results section.

#### 2.4.2. Secondary Outcomes


(1)Numeric pain rating scale (NPRS)


The NPRS is an 11-point numeric pain intensity scale ranging from 0 (“no pain”) to 10 (“as much pain as possible or intolerable pain”). A change of 2 points or more was identified as the minimal clinically important difference (MCID) in participants with chronic neck pain [33].
(2)Neck disability index (NDI)

The NDI is a patient-completed, condition-specific functional status questionnaire with 10 items. The total score of this questionnaire ranges between 0 and 50 points, with higher scores indicating higher levels of disability, which is expressed as a raw score with a maximum score of 50. The MCID of the NDI is 5.5 [20,33,34].
(3)Active cervical range of motion (CROM)

Active CROM was measured in a sitting position using a CROM goniometer (CROM Deluxe model; Performance Attainment Associates, Roseville, MN, USA). The CROM allows measurement of ROM in three planes (flexion/ extension, lateral flexion, and rotation about gravity). Participants sat upright and were asked to move their necks in each direction 3 times. Documentation of cervical ROM was expressed in the form of full range, a total value for the sagittal (flexion and extension), frontal (lateral flexion right and left), or transverse plane (rotation right and left), in the form of 3 measurements [35,36].
(4)Three-dimensional posture parameters of the head in relation to the thoracic region

A 3D postural assessment, known as the Global Posture System (GPS) 600, (Chinesport, Udine, Italy), was used to examine the postural displacement variables [37,38,39]. This device was used per the manufacturer’s instructions [37]. The device has a unit for podoscopic analysis, a unit for postural analysis, and a stability measuring platform, and it comes with an image acquisition system and custom software. The camera of the image acquisition system was positioned 107 cm from the ground and 190 cm from the subject. The reliability and validity of this device have been verified previously where measurements demonstrated excellent within-rater reliability (ICC = 0.89) and standard error of measurement (SEM) = 1.5 degrees with a minimum detectable change (MDC) = 1.9 degrees; while inter-rater reliability is good to excellent (ICC= 0.7) [40,41]. We analyzed the posture of the head in relation to the thoracic region in terms of translations and rotations.

#### 2.4.3. Assessment Procedures


A.Preparation of patients:


The patients were asked to wear tight-fitting clothes to allow the examiners to find various anatomical sites. The examiners placed 13 markers on each patient before taking the four photographs.
B.Marker placement:

Antero-posterior and lateral view marker locations are shown in Figure 1. The points over which the markers were fixed were well-cleaned with alcohol to remove any moisture and to ensure good fixation. Four photographs or four views were obtained for every patient, one anterior and one posterior view and two lateral (right, Rt, and left, Lt) views.
C.Starting position of the patients:

For the photographs, patients were instructed to stand on the lux postural analyzer part of the GPS, to take a deep breath 3–5 times for full relaxation, to nod their head up and down twice with their eyes closed, and to assume what they felt to be a neutral body posture then participants’ eyes were opened, and they remained still, without any motion, during this stance. Four digital photographs were taken using a computer mouse. The set of photographs was processed through secure software analysis using GPS.

#### 2.4.4. Measured Items (the Postural Parameters) of the Head Region in Relation to the Thoracic Region

A right-handed Cartesian coordinate system with x-axis positive to the left, y-axis positive vertically, and z-axis positive to the anterior was used to describe postures of the head as translation displacements in centimeters (Tx, Ty, and Tz) along these axes and, in addition, as rotation displacements (Rx, Ry, and Rz) in degrees from a normal upright stance. Vertical translations (Ty), which would require radiographic analysis of hypo- or hyper-lordosis, were not calculated in the present study as is shown in Figure 2 [23].

#### 2.4.5. Postural Translations of the Head in Relation to the Thoracic Region

Tx (Rt. or Lt. side shifting or lateral translation), is the measure of the horizontal distance from the vertical line passing through the middle sternal notch to the vertical line passing through the nose [16].

Tz (anterior head translation), is the measure of the horizontal distance from the vertical line crossing the middle acromion process to the vertical line crossing the tragus of the ear [22].

#### 2.4.6. Postural Rotations of the Head in Relation to the Thoracic Region

Rx (flexion or extension position of upper cervical), is the measurement of the angle between the tragus of the ear, the canthus of the eye, and the horizontal line [10].

Ry (Rt. or Lt. rotation), is the measurement of the angle between the glabella of the forehead or tip of the nose, the middle point of the chin, and the vertical line [23].

Rz (Rt. or Lt. side bending), is the measurement of the angle between the inferior margins of the right and the left ear and the horizontal line [10].

### 2.5. Interventions

Both groups received conventional or local treatment consisting of a moist hot pack, soft tissue mobilization, manual therapy, and therapeutic exercises [42,43,44,45] (Table 1). Only the study group received 3D PCO to perform the mirror image therapy (reverse posture training) for 20–30 min while the patient was walking on a motorized treadmill 2–3 times per week for 10 weeks. The CONSORT flow chart diagram for this trial is presented in Figure 3.

#### Study Group 3D PCO Performed the Mirror Image^®^ Therapy (Reverse Posture Training) While the Patient Was Walking on Motorized Treadmill

Mirror image therapy (reverse posture training) was delivered via the use of the adjustable PCO (patent number CN201921929736.1), as shown in Figure 4. The PCO was applied, to reverse the abnormal posture according to the 3D posture analysis data, while the patient was walking at approximately 2–3 miles per hour on a standard motorized treadmill for 20–30 min per session. To facilitate tissue remodeling and to stretch ligamentous tissues reverse posture training was applied in the mirror image traction or therapeutic position; an example of participant data is shown in Table 2. Then, walking training using the treadmill was performed during which the participant’s mirror image traction could be held by the adjustable PCO based on mirror image therapy, which has been previously used in other studies based on Harrison et al., 2004 [16,17,46]. The rationale for walking on the treadmill while maintaining a patient’s mirror image position is based on the concept of neuromuscular retraining of motor patterns that have developed over time in both static and dynamic posture tasks and is based on the earlier randomized trial by Diab and Moustafa [17]. This program was repeated 2–3 times/week for 10 weeks, as in Figure 5 (as shown in the Appendix A).

Participants in both groups attended 30 physical therapy treatment sessions over a 10-week period at 2–3 sessions per week. Short-term follow-up evaluations were performed after 10 weeks of interventions. To minimize therapist variation and to increase consistency, the same physiotherapist independently delivered the entire intervention program for every participant. Every physiotherapist had 10 years of experience and received training for the application of the specific interventions for one week before starting the study. Participants in both groups were advised to perform all therapeutic exercises once daily as their home routine during non-treatment days and to follow the posture correction advice. Record sheets were collected every week and were subsequently analyzed to calculate the mean exercise frequency per week and the mean exercise time per day. To monitor the exercise times and the number of sets performed during the study accurately, videos of the exercises, photos of postural correction, and a record sheet were distributed to the participants.

### 2.6. Statistical Analysis

Primary outcome measures and their results described in our study protocol [47] are discussed descriptively in Table 3. To estimate the sample size for a future full-scale RCT, between-group effect sizes and 95% confidence intervals (CIs) with Hedges’ correction were calculated for the change in the secondary outcomes of NPRS, NDI, active cervical ROM, and 3D posture parameters. The mean ± standard deviation (SD) value for each of the secondary outcomes was used in the calculation of the effect size. The estimated sample size was then determined using the between-group effect size with a minimum of 80% power (α = 0.01 or 3D posture parameters and α = 0.05 for other secondary outcomes) using G power software. The sample size will be increased by 20% to allow for an estimated dropout rate in the future RCT. Statistical methods for the secondary outcome measures were evaluated by comparing the change within groups (from baseline to post-treatment) and then estimating the within-group effect size. Complete analyses were conducted to include outcomes from all participants who completed baseline and post-treatment evaluations as recommended in CONSORT guidelines [48]. The between-group difference in change scores for each outcome measure from baseline to post-treatment was calculated as the mean and 95% CI. All statistical analyses were performed with SPSS Version 2.2 software (IBM Corporation, Armonk, NY, USA). Correlations (Pearson’s r) were used to examine the relationships between the 3D postural parameters and all measured outcomes.

## 3. Results

Of 54 people who responded to advertisements in orthopedic and rehabilitation department clinics and were interested in participating in the study, 11 people did not have time to complete the intervention, 9 people had NPRS less than 3 and NDI less than 5, and 10 people did not meet the inclusion criteria of 3D posture measurements of the head in relation to the thoracic region. In the end, 30 people were excluded and 24 people were included (45% who fulfilled inclusion criteria and 100% of included participants enrolled in the study), as shown in Figure 3. Those 24 participants fulfilled all procedures of assessment and interventions, but 3 (12.5%) participants were lost and did not make the final assessment (2 from the control group and 1 from the study group). Complete demographic characteristic data of participants are shown in Table 4.

The results of the primary outcome aspects related to feasibility are provided in Table 3. Using the extension of the CONSORT statement for pilot and feasibility studies when developing the protocol our findings were informed by our previously published protocol. The recruitment into our study was achieved in an acceptable period, and less than 15% of participants were lost to follow-up before the final assessment because they had to travel to other cities and did not have time to continue the treatment and assessment. We used complete case analyses, where 12.5 % (3 of 24) of participants were excluded because of missing data.

Table 5 shows the within-group change in each of the secondary outcome measures for each group. In general, larger improvements and greater effect sizes were found in the study group for pain, disability, postural measures, and cervical spine range of motion.

Between-group differences in change scores, effect size, and estimated sample size for each of the secondary outcome measures of NDI, NPRS, and range of motion are shown in Table 6. While Table 7 presents the between-group differences in change scores, effect size, and estimated sample size for each of the secondary outcome measures of 3D posture displacements of Tx, Tz, Rx, Ry, and Rz. In general, larger improvements and greater effect sizes were found in the study group for pain, disability, postural measures, and cervical spine range of motion in Table 6 and Table 7. Sample size estimates for the full-scale RCT indicated that a minimum of 14 participants (ROM for flexion and extension) and a maximum of 80 participants (ROM y-axis rotation) would be needed for full statistical evaluation. A full-scale RCT using our multimodal program for participants with CNSNP related to poor posture would require a sample of 42 participants (without calculating any dropout) to demonstrate a clinically meaningful functional improvement based on the NDI. Table 6.

We found a moderately positive correlation between pre- and post-treatment changes in 3D postural parameters and pre- and post-treatment changes in pain and NDI, indicating that as posture displacement decreased in our population, so did pain intensity and NDI scores. However, as shown in Table 8, we discovered a moderately negative correlation between cervical ROM values and pre- and post-treatment changes in 3D postural parameters.

## 4. Discussion

Our promising results suggest that it is feasible to conduct a full-scale RCT using a 3D PCO to perform mirror image therapy (reverse posture training) while a patient is walking on a motorized treadmill. Based on our data, a full-scale RCT using our multimodal program for participants with neck pain related to poor posture or postural neck pain would require a sample of 42 participants (without calculating any dropout) to demonstrate a clinically meaningful functional improvement based on the NDI. While feasible, our results also suggest that some modifications to the protocol may enhance participant enrolment, including access to the intervention and the effectiveness of various aspects of the intervention in future studies. Furthermore, since study participants indicated that the treatment sessions were quite lengthy, a reduction in treatment time is needed in a future full-scale trial. This can be accomplished by reducing the number of interventions (hot packs and one of the mobilization procedures) and reducing the walking time on the treadmill to 15–20 min instead of 20–30 min.

In our pilot study, 54 people responded to advertisements, 24 (45%) of whom were eligible. Therefore, at least 93 potential participants would be required to respond to advertisements to obtain a sample of 42 participants. This information will assist in planning the extent of the intervention, timelines for recruitment, and budgets for future studies.

The diagnosis was based on physical examination, including history, demographic variables, the mode of onset, duration of symptoms, nature, and location of symptoms, as well as questions regarding aggravating and relieving factors, such as posture modifications and change positions and any prior history of neck pain (3). The assessment also depended on pain level, neck disability, 3D posture analysis of the head in relation to the thoracic region, and active cervical range of motion. We subsequently used an X-ray to exclude any specific cause of pain. Notably, a large number (30/54, 55%) of participants were excluded; they had neck pain, but their NPRS was less than 3 and NDI was less than 5, and posture modifications or poor posture were not the risk factor for the problem occurring. Those symptoms were diagnosed as myofascial pain syndrome. Other participants had NPRS and NDI and a score of more than 3 and 5, respectively, but did not match the 3D posture analysis criteria; the cause of the problem was not due to posture modifications or poor posture. We tried to include participants who had poor posture according to the 3D analysis, and to modify poor posture, which affects participant symptoms and function; ultimately, we attempted to include only participants with postural components to their neck pain [10,13].

We included neck pain and a disability score from moderate to severe on the NPRS and NDI. Most participants had a fixed position during smartphone use or in the workplace for long periods or had a monotonous constrained vision-related task such as computer programming. Only three participants were housewives using smartphones for extended times in the flexed neck position; the other participants were university students and desk office workers who were often using computers in a slumped seated position in FHP for a prolonged time.

To the best of our knowledge, this intervention was the first to use a 3D PCO for mirror image therapy (reverse posture training) of the cervical spine while the patient is walking on a motorized treadmill and to utilize a supervised, tailored brace for each participant according to 3D posture analysis, as well as functional walking training for at least 20 min using the PCO as active not passive therapy. Within-group effect sizes for improvement of 3D posture analysis data of the head, in relation to the thoracic region (five posture variables, two translation displacements, and three rotational displacements), in the study group were very large (1.19–1.87). Interestingly, the control group also had small to medium gains in 3D posture analysis data (effect size, 0.15–0.38). Likely speaking, the larger changes in the study groups’ postures are due to the targeted mirror image therapy using the PCO with functional walking training on a treadmill. The training was thus tailored according to each participant’s 3D analysis data. Because both groups practiced a therapeutic exercise program as shown in Table 1, this might explain the posture improvement in some parameters (Rx, Ry, and Rz) of 3D posture analysis in the control group. In addition, the study group reported large within-group improvements in pain, function, and quality of life, but due to the limitation in the sample size, we can only infer that this was due to the PCO training and postural correction.

Future fully powered RCTs should explore whether greater improvements in pain and quality of life are associated with improvements in 3D posture parameters and whether or not these continue to improve after follow-up for 3 to 6 months or longer. Our future RCT will include these specifications, especially a follow-up period of 6 months (ClinicalTrials.gov ID: NCT04263883), and will use a sham or a placebo brace for the control group.

In our pilot study, the effect size for between-group differences in change scores is moderately large in the study group for all secondary outcomes, such as NDI, NPRS, and all active neck ROM, as well as all parameters of 3D posture analysis. The changes visualized on photographic measurements would be due to the application of traction forces to the lateral cervical structures or the reverse posture traction (reverse posture training) on 3D planes. The muscles and ligamentous structures of the spine are viscoelastic. The deformation of these structures is, mechanically, time-dependent and force-dependent [16]. When under loading, spinal ligaments complete a stress relaxation process in approximately 500 s (8.33 min). However, the intervertebral disc will continue to deform for 20 min to 60 min [16,49,50]. For this reason, we progressively increased the PCO therapy up to 20 min in the form of functional training, such as walking, to attain the maximum amount of deformation to the paraspinal structures in a clinically efficient time.

The strengths of our study included using the extension of the CONSORT statement for pilot and feasibility studies when developing the protocol [48,51]. In addition, our findings were informed by our previously published protocol [47]. The recruitment into our study was achieved in an acceptable period, and less than 15% of participants were lost to follow-up before the final assessment because they had to travel to other cities and did not have time to continue the treatment and assessment. We used complete case analyses, where 12.5% (3 of 24) of participants were excluded because of missing data. Despite options for the statistical imputation of missing data, minimizing the dropout rate should be a priority in our future studies.

## 5. Limitations

The study had some potential limitations, each of which points toward directions for future study. The first limitation of our study was the lack of blinding of participants and physiotherapists because of the nature and difference of the interventions. It was difficult to blind participants and healthcare providers. However, the investigator, outcome assessor, and data analyst were blinded to the participant allocation group. We can overcome this issue in future studies by adding a placebo-treated group for mirror image traction using another orthotic intervention, such as another cervicothoracic brace without adjustment according to 3D posture analysis. Additionally, our study results are limited to the outcome measurements chosen to evaluate CNSNP. It is possible that using different outcome measures of neck pain, such as muscle endurance, motor control, and proprioceptive tasks, would produce different findings between the intervention and control groups. Third, the assessment of psychosocial models, such as depression and fearful avoidance of movement, were not included in our pilot, and we will assess them in a future RCT. Fourth, our study did not include a true natural history group with chronic neck pain and participants must not have had physical therapy treatment in the previous 6 months, but it was unknown which other treatment they might have received prior to the previous 6 months.

The last limitation was the small sample size, which weakens any strong interpretation regarding the effectiveness of our intervention. We could not solve all these limitations at the same time in one study because the PCO used in the study is a new device for therapeutic use. Instead, we focused on the design of a new postural mirror image brace and first assess its feasibility, the pilot study outcomes, and the secondary assessment of its direct effect on 3D posture parameters and its indirect effect on neck pain, disability, and active neck ROM.

## 6. Conclusions

It was demonstrated that a full-scale RCT of a 3D PCO to perform mirror image therapy (reverse posture training) is feasible. Adding a 3D PCO to a multimodal program positively affected neck pain management outcomes by reducing neck pain, improving neck function, and increasing active ROM, which was likely due to improved 3D posture alignment of the head. Adequately powered and improved studies are needed to confirm or refute this association.

## Figures and Tables

**Figure 1 jcm-11-07028-f001:**
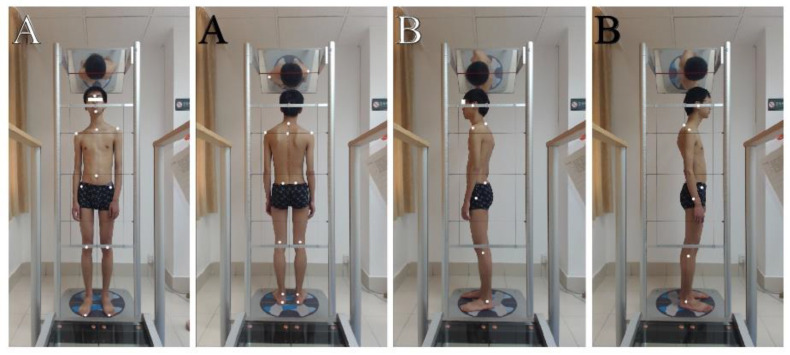
The photographs taken using the Global Posture System (GPS). (**A**) Anterior and posterior views. (**B**) Sagittal plane or lateral views. The six reflective markers used in the analysis are: acromion, anterior superior iliac spine, posterior superior iliac spine, glabella, tragus, C7, and middle sternal notch.

**Figure 2 jcm-11-07028-f002:**
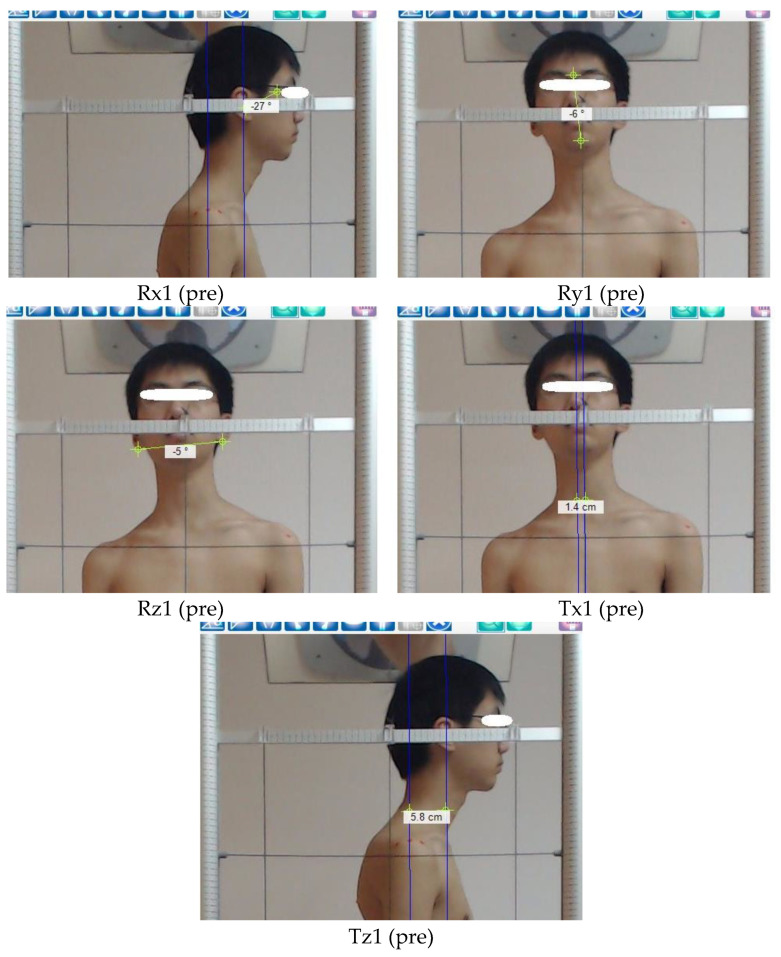
Three-dimensional postural parameters of the head region in relation to the thoracic region. Postural rotations (**Rx**, **Ry**, **Rz**). Postural translations (**Tx**, **Tz**).

**Figure 3 jcm-11-07028-f003:**
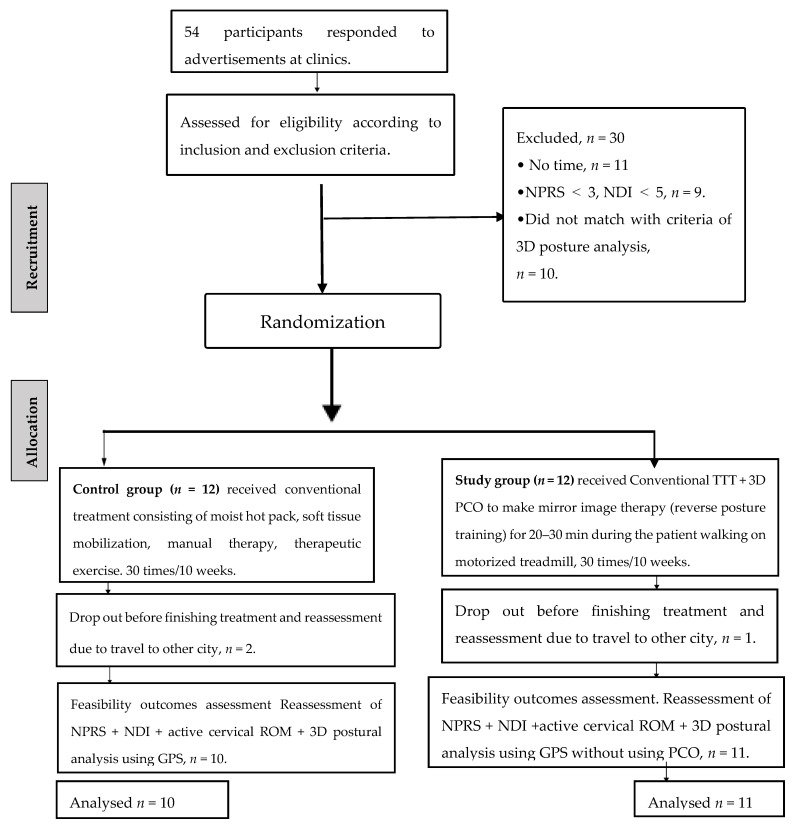
The CONSORT flow chart diagram for the trial.

**Figure 4 jcm-11-07028-f004:**
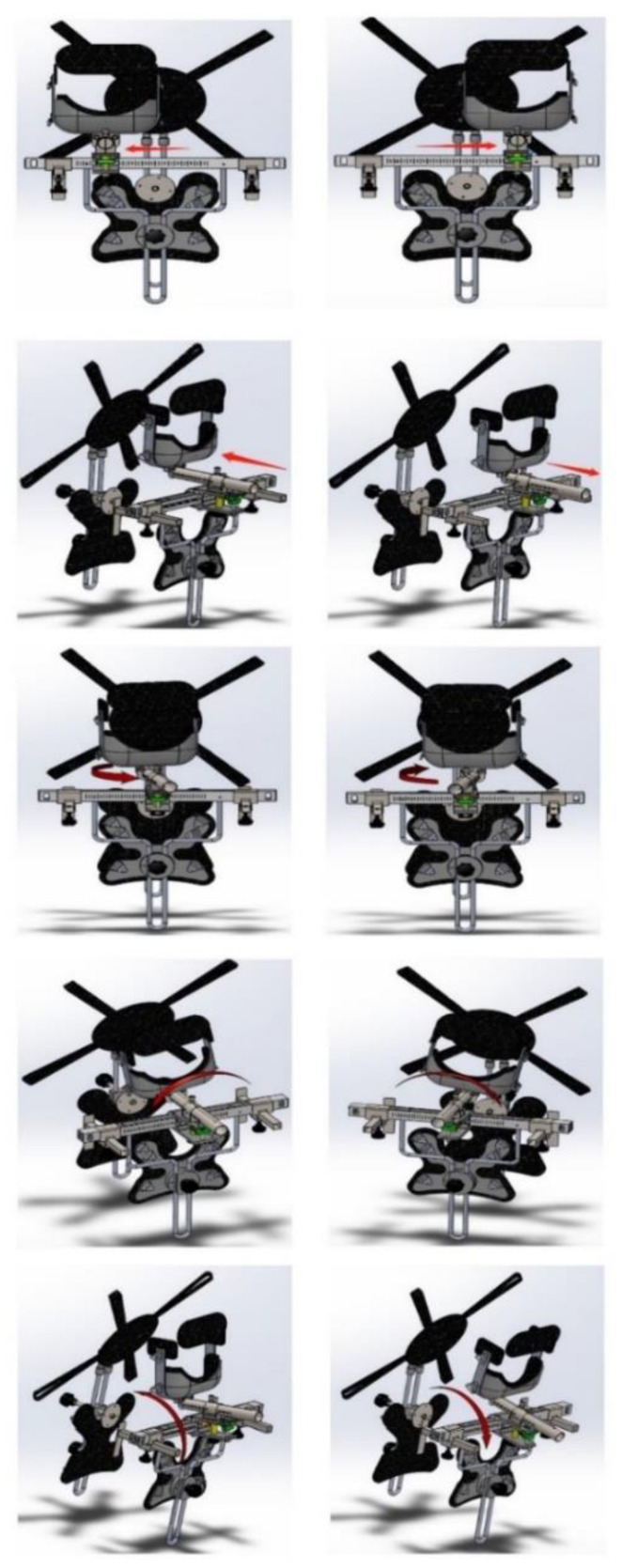
Posture corrective orthotic (PCO) demonstrating the availability to move in different directions and lock the head in its opposition position. Top row: lateral translation of the head left and right (Tx). Second row: anterior and posterior translation of the head (Tz). Third row: rotation of the head about vertical gravity left and right (Ry). Fourth row: side bending of the head left and right (Rz). Bottom row: extension and flexion of the head (Rx). See Figure 5 for a sample patient setup.

**Figure 5 jcm-11-07028-f005:**
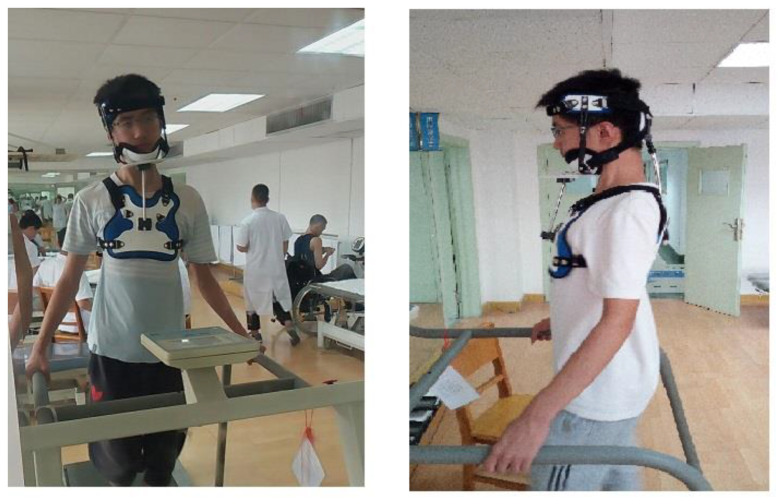
Three-dimensional PCO to perform the mirror image therapy (reverse posture training) while the patient was walking on motorized treadmill.

**Table 1 jcm-11-07028-t001:** Both the study and control groups received conventional or local treatment consisting of a moist hot pack, soft tissue mobilization, manual therapy, and therapeutic exercises. Description of this conventional treatment, exercise prescription, and progression are presented herein. Participants in both groups attended 30 physical therapy treatment sessions over a 10-week period at 3 sessions per week. See methods section for complete details of progressions.

Conventional Treatment	Description
**Moist hot pack**	Applied to the area of pain at neck region muscles, such as the upper part of the trapezius, levator scapulae, splenius capitis, and cervicis muscles, for 15 min.
**Soft tissue mobilization**	Deep stroking massage performed along the entire length of the taut band within the painful or tight muscles.
**Cervical mobilization**	Low-velocity passive mobilization techniques to the symptomatic cervical segments as determined by the physiotherapist’s clinical examination. Physiotherapists could be able to select from what were termed passive accessory and physiological movement techniques as believed appropriate to the individual participant based on the initial and progressive reassessments.
**Therapeutic exercises** Cervical flexors	Level 1 Re-education of CCF movement pattern 1. Supine, knees bent -Gentle and controlled nodding action facilitated with eye movement 10 reps Holding. 2. Supine, knees bent -Repeated and sustained CCF 10 s holds × 10 reps Level 2 Interaction between the deep/superficial cervical flexors 1. Sitting -Controlled head movement through range of extension and return to neutral 10 reps Co-contraction of the deep cervical flexors/extensors. Level 3 Strength/endurance of the cervical flexors 1. Sitting -Isometric CCF in a range of cervical extension 10 s holds × 10 reps -Lifting the head off the wall (with the chair up to 30 cm away from the wall) 10 s holds × 10 reps 2. Supine -Lifting the head off a pillow (2 or 1 then 0 pillows as per participant’s capacity) 10 s holds × 10 reps.
Cervical extensors	Level 1 Re-education of extension movement pattern 1. Prone on elbows/four-point kneeling positions -Cranio cervical extension 3 sets of 5 reps. -Cranio cervical rotation (<45°) 3 sets of 5 reps. -Cervical extension while keeping the cranio cervical region in a neutral position 3 sets of 5 reps. Level 2 Co-contraction of the deep cervical flexors/extensors 1. Sitting -Isometric cervical rotation facilitated with eye movement (left/right sides) 5 s holds × 5 reps. Level 3 Strength/endurance of the cervical extensors 1. Prone on elbows/four-point kneeling positions -Isometric hold in range of cervical extension 10 s holds × 10 reps.
Cervico Scapular control	Level 1-Re-education of scapular movement control Cervico scapular muscle control 1. Sitting -Arm movement without load (external rotation/abduction/flexion < 30°) 10 reps -Arm movement without load throughout range 10 reps 2. Prone on elbows/four-point kneeling position -Thoracic lift (serratus anterior) and isometric hold 5 s holds × 5 reps. Level 2 Strength/endurance of cervico scapular muscles 1. Sitting -Arm movement with load using Thera-band (external rotation/abduction/flexion < 30°) 10 reps. -Arm movement with load throughout the range 10 reps. 2. Prone -Lift the shoulder off the bed and hold without arm load 10 s holds × 10 reps. -Lift the shoulder off the bed and hold with arm load using Thera-band 10 s holds × 10 reps.
Postural correction	Level 1 Correction of spinal posture Sitting -Active upright sitting initiated with lumbo-pelvic movement 10 s holds × 10 reps Level 2 Correction of spinal posture and scapular orientation Sitting -Actively positioning the scapular in a neutral posture while maintaining spinal posture 10 s holds × 10 reps Level 3 Spinal and scapular correction plus occipital lift Sitting -Actively lengthen the back of the neck while maintaining spinal and scapular posture 10 s holds × 10 reps. Standing on wall Actively extend spine then chin in cervical with squeezing abdomen 10 s holds × 10 reps.

CCF: craniocervical flexion; reps: repetitions.

**Table 2 jcm-11-07028-t002:** Example of 3D posture analysis data and mirror image therapy (reverse posture training) using the posture corrective orthotic PCO.

3D Posture Analysis of Head in Relation to Thoracic	Reverse 3D Posture Data by PCO (Mirror Image Therapy)
1. Rx (extension position of the head) = 25° − 18° = 7° extension.	7° flexion of the head.
2. Ry (right or left rotation of the head) = 6° left rotation.	6° right rotation of the head.
3. Tz (anterior head translation = 5.8 cm anterior head translation.	5.8 cm posterior head translation.
4. Rz (right or left side bending) = 5° right side bending.	5° left side bending.
5. Tx (side shifting of the head = 1.4 cm right side shifting.	1.4 cm left side shifting.

**Table 3 jcm-11-07028-t003:** Results of primary aim (feasibility).

Primary Aim/Criteria	Description
Integrity of the study protocol • Recruitment: minimum requirement of 80% of eligible participants entering study.	With 45% of interested participants being eligible. 100% of eligible participants enrolled in the study. Inclusion criteria of the study protocol appeared acceptable.
Validity of eligibility criteria	The eligibility criteria were followed with another published paper [19], and our protocol that was published previously [47].
Understanding and integrity of intervention for treating physical therapists	During the post-study interview, physical therapists said that one-week training before study was appropriate and enough, continuous communication with the study authors was essential to ensure that the protocol was followed throughout the study.
Convenience of intervention for participants	During the post-study interview, participants upraised the concern that with one treating physical therapist, appointments availability were limited. They felt that more availability of appointments would enhance recruitment and retention or more than treating physical therapist for every participant.
Integrity and suitability of intervention to participants	During the post-study interview, participants in both groups said that they believed the intervention was valuable, and they would participate in the study again. Only one issue was that the treatment period was too long and, hope to reduce treatment period of intervention in the future, and none expressed offense about being randomized to the control intervention.
Feasibility of study time requirement and study facilities for participants	During the post-study interview, while participants stated that a large time commitment was required to participate in the study, they all acknowledged that this was necessary for improvement. However, they advised us to reduce treatment sessions to less than 30 times although ten weeks was good time to make improvement in posture deviations. All participants felt that the services in which the interventions were delivered were appropriate.
Recruitment and retention procedures • Goals for minimum requirement for adequate recruitment and retention: at least 80% of participants attended 75% of appointments and completed 75% of the prescribed exercises	Fifty-four people responded to recruitment at rehabilitation clinic over a 9-months inclusion period. Of these, 45% fulfilled inclusion criteria, and all were included in the study at rehabilitation clinic. Of the 24 participants, 21 attended all treatment sessions and 3 lost before finishing treatment. No adverse events were recorded. Exercise intervention compliance was measured using either record sheet diaries or via WeChat application.
Testing of outcome measurement collection • Determined by completeness of outcome data collected, and through post-study interview	At the end of the study, all 21 participants who finished the study completed patient-reported outcome questionnaires, neck ROM and 3D posture parameters. The blinded outcome assessor, with no missing data, collected it for all participants.
Suitability of randomization procedure and methods used to ensure blinding • Determined during post-study interviews of treating physical therapist, blinded outcome assessor. Selection of the most appropriate primary outcome measure for a full-scale RCT Determined based on outcome measure with largest between group effect size	The randomization procedure was appropriate, with the treating physical therapist informed of group allocation but outcome measurement assessor was not aware of group allocation of any participants. All participants knew group allocation because of heterogeneity of interventions. NDI (0.79), as a measure of neck disability and quality of life, and 3D posture parameters of head in relation to thoracic (Tx, Tz, Rx, Rz, Ry) were selected.
Estimation of required sample size for a fully powered study • Based on NDI	A sample size of 42 participants (21 in each group) provides a minimum of 80% power (α = 0.05) and is required for an effect size of 0.79. To account for an estimated 20% dropout, the recommended sample size is 50 participants (25 in each group).

NDI: neck disability index; ROM: range of motion; 3D: three-dimensional.

**Table 4 jcm-11-07028-t004:** Demographic characteristic data of participants.

	Study Group (*n* = 12)	Control Group (*n* = 12)	*p*-Value
Age (y) mean ± SD	27.4 ± 5.5	27.2 ± 4.7	0.7
Weight (kg) mean ± SD	64.58 ± 6.9	67.2 ± 5.8	0.3
Height (m) mean ± SD	1.65 ± 0.5	1.67 ± 0.6	0.6
BMI (kg/m^2^), mean ± SD	23.5 ± 1.2	24.2 ± 0.97	0.2
Male, *n* (%)	8 (66.7%)	7 (58%)	0.5
Female, *n* (%)	4 (33.3%)	5 (42%)	0.5
Participants Employment			
University student, *n* (%)	7 (58%)	6 (50%)	0.6
Desk office worker *n* (%)	4 (33.3%)	4 (33.3%)	0.5
House wife, *n* (%)	1 (8.3%)	2 (16.7%)	0.6
Married, *n* (%)	4 (33.3%)	3 (25%)	0.7
Duration of Pain, *n* (%)			
3–24 months	8 (66.7%)	9 (75%)	0.6
>24 months	4 (33.3%)	3 (25%)	0.7
Current use of Medications, *n* (%)			
Yes	2	3	0.6
No	10	9	0.7
Referred pain, *n* (%)	6 (50%)	7 (58%)	0.5
Current smoker, *n* (%)	2 (16.7%)	3 (25%)	0.6

Mean ± SD: standard deviation; BMI: body mass index.

**Table 5 jcm-11-07028-t005:** Within-group change in secondary outcome measures for each group. Postural translations (Tx and Tz) are measured in centimeters, postural rotations (Rx, Ry, Rz), and ranges of motion (ROM) are measured in degrees. The mean difference (MD) is the difference between the baseline and after-treatment values. A negative value for the change (MD) in range of motion indicates an increase in the overall motion for the variable.

	Baseline	Post Intervention	MD ^†^ (*p* Value 95% CI)	ES (*d*)
	Mean ± SD	Mean ± SD		
**NDI (0–50)** **Study G**	12.42 ± 4.54	3.33 ± 2.42	9.1 ± 4.3 <0.001 * (6.4, 11.8)	2.5
**Control G**	12.41 ± 3.02	5.41 ± 2.35	7 ± 1.13 <0.001 * (6.4, 7.7)	2.5
**NPRS (0–10)** **Study G**	5 ± 1.4	1.8 ± 1.03	3.2 ± 1.26 <0.001 * (2.4, 3.9)	2.6
**Control G**	4.91 ± 1.2	2.29 ± 0.87	2.6 ± 0.4 <0.001 * (2.4, 2.9)	2.49
**Tx Study G**	0.97 ± 0.4	0.41 ± 0.13	0.56 ± 0.37 <0.001 * (0.3, 0.8)	1.87
**Tx Control G**	0.75 ± 0.35	0.68 ± 0.31	0.075 ± 0.05 0.29 (0.04, 0.1)	0.2
**Tz Study G**	3 ± 1.3	1.34 ± 1.1	1.6 ± 1.05 <0.001 * (0.9, 2.4)	1.2
**Tz Control G**	3.1 ± 1.4	2.85 ± 1.6	0.24 ± 0.28 0.23(0.06, 0.4)	0.15
**Rx Study G**	24.8 ± 4.17	19.41 ± 2.54	5.4 ± 3.2 <0.001 * (3.4, 7.45)	1.56
**Control G**	23.8 ± 3.95	22.3 ± 3.98	1.5 ± 2.3 0.013 (0.03, 2.96)	0.38
**Ry Study G**	3.5 ± 1.6	1.58 ± 0.67	2 ± 1.88 <0.001 * (0.7, 3.1)	1.35
**Ry Control G**	3 ± 1.3	2.33 ± 1.4	0.5 ± 0.52 0.16 (0.2, 0.8)	0.29
**Rz Study G**	3.3 ± 1.5	1.5 ± 0.79	1.83 ± 1.8 <0.001 * (0.7, 2.97)	1.34
**Rz Control G**	3 ± 1.4	2.33 ± 1.4	0.5 ± 0.52 0.2 (0.2, 0.8)	0.29
**ROM flex and extension** **Study G**	87.1 ± 4.18	108.1 ± 4	−21 ± 0.18 <0.001 * (−21.3, −20.7)	5
**Control G**	86.7 ± 4.37	100.6 ± 4	−13.9 ± 0.37 <0.001 * (−14.4, −13.5)	3.3
**ROM lateral flexion** **Study G**	65.75 ± 4.5	80.25 ± 4	−14.5 ± 0.5 <0.001 * (−14.7, −14.3)	3.4
**Control G**	63.91 ± 4.6	73.41 ± 4.2	−9.5 ± 0.4 <0.001 * (−9.8, −9.4)	2.1
**ROM rotation** **Study G**	101.8 ± 2.33	118.8 ± 2	−17 ± 0.33 <0.001 * (−17.1, −16.96)	7.8
**Control G**	101.58 ± 2.2	117.48 ± 2	−15.9 ± 0.02 <0.001 * (−16, −15.8)	7.5

Mean. MD, mean difference NDI, neck disability index. NPRS, numeric pain rating scale. ROM, range of motion. Flex. flexion. ES (*d*), effect size (Cohen’s *d*), Tx, side shifting of head. Tz, Ant. head translation. Rx, upper extension of head. Ry, R.t or l.t rotation of head. Rz, side bending R.t or l.t of head. ^†^ Values in parentheses are 95% confidence interval. G, group. Sig.; significant, 001, Sig; 0.01. * Significant for TZ, TX, Rx, Ry, Rz. Sig; 0.05, * significant for all other outcomes.

**Table 6 jcm-11-07028-t006:** Between-group differences in change scores, effect size, and estimated sample size for each of the secondary outcome measures with significance level of 0.05. The mean difference (MD) is the difference between the two groups’ change score values. A negative value for the change (MD) in range of motion indicates an increase in the overall motion for the variable for that group whereas the difference between the group is a positive number indicating greater improvement for the study group. Ranges of motion (ROM) are measured in degrees.

	Study Group Change Score * (Baseline to Posttreatment)	Control Group Change Score * (Baseline to Posttreatment)	MD ^†^ (*p* Value 95% CI)	ES (*d*)	Estimated Total Sample Size for Outcome Measure, *n* ^‡^
**NDI (0–50)**	9.1 ± 4.3	7 ± 1.13	2.08 <0.001 * (4.11, 0.06)	0.79	42
**NPRS (0–10)**	3.2 ± 1.26	2.6 ± 0.4	0.458 <0.001 * (1.26, 0.35)	0.58	76
**ROM flex and exten.**	−21 ± 0.18	−13.9 ± 0.37	7.42 <0.001 * (3.79, 11.1)	1.78	14
**ROM lateral flexion**	−14.5 ± 0.5	−9.5 ± 0.4	6.8 <0.001 * (2.97, 10.7)	1.26	22
**ROM rotation**	−17 ± 0.33	−15.9 ± 0.02	1.35 <0.001 * (−0.49, 3.2)	0.56	80

NDI, neck disability index. NPRS, numeric pain rating scale. ROM, range of motion. Flex and exten., flexion and extension. MD, mean difference. * Values are mean ± SD. ^†^ Values in parentheses are 95% confidence interval. ^‡^ Estimated sample size determined using Student *t* test sample size calculation, without adjusting for anticipated dropouts and losses to follow-up (α = 0.05, β = 0.80). Sig; 0.05. ES (*d*), effect size (Cohen’s *d*).

**Table 7 jcm-11-07028-t007:** Between-group differences in change scores, effect size, and estimated sample size for 3D posture parameters of head related to thoracic with significance level 0.01. Postural translations (Tx and Tz) are measured in centimeters and postural rotations (Rx, Ry, Rz) are measured in degrees.

	Study Group Change Score *	Control Group Change Score *	MD ^†^ (*p* Value 95% CI)	ES (*d*)	Estimated Total Sample Size for Outcome Measure, *n* ^‡^
**Tx (side shifting of head)**	0.56 ± 0.37	0.07 ± 0.04	.48 <0.001 * (−0.48, −0.05)	1.70	18
**Tz (Ant. H. Translation)**	1.6 ± 1.05	0.24 ± 0.28	1.36 <0.001 * (−2.6, −0.28)	1.1	38
**Rx (upper extension of head)**	5.4 ± 3.2	1.5 ± 2.3	3.9 <0.001 * (−5.75, −0.09)	1.12	36
**Ry (rot. R.t or l.t of head)**	2 ± 1.88	0.5 ± 0.52	1.5 <0.001 * (−1.72, 0.22)	0.96	48
**Rz (side bending R.t or l.t of head)**	1.83 ± 1.8	0.5 ± 0.52	1.33 <0.001 * (−1.8, 0.15)	0.90	52

H, head. rot., rotation. * Values are mean ± SD. ^†^ Values in parentheses are 95% confidence interval. ^‡^ Estimated sample size determined using Student *t* test sample size calculation, without adjusting for anticipated dropouts and losses to follow-up (α = 0.01, β = 0.80). Sig; 0.01. ES (*d*), Effect size (Cohen’s *d*).

**Table 8 jcm-11-07028-t008:** Correlations (Pearson’s *r*) were used to examine the relationships between the 3D postural parameters and all measured outcomes for the entire sample. * Indicates a statistically significant difference at *p* < 0.001.

	Change in Neck Pain	Changes in NDI	Changes in ROM Flex and Exten	Changes in ROM Lateral Flexion	Changes in ROM Rotation
**Change in Tx (side shifting of head)**	0.5 <0.001 *	0.5	−0.48 <0.001 *	−0.51 <0.001 *	−0.41 0.01
**Change in Tz (Ant. H. Translation)**	0.6 <0.001 *	0.31 0.06	−0.53 <0.001 *	−0.43 0.04	−0.33 0.05
**Change in Rx (upper extension of head)**	0.3 0.06	0.49 <0.001 *	−0.51 <0.001 *	−0.32 0.05	−0.3 0.06
**Change in Ry (rot. R.t or l.t of head)**	0.5 <0.001 *	0.62 <0.001 *	−0.4 0.01	−0.56 <0.001 *	−0.71 <0.001 *
**Change in Rz (side bending R.t or l.t of head)**	0.4 0.01	0.46 <0.001 *	−0.3 0.06	−0.51 <0.001 *	−0.54 <0.001 *

## Data Availability

The datasets analyzed in the current study are available from the corresponding author upon reasonable request.

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
