# Peer review of "Randomized Feasibility Pilot Trial of Adding a New Three-Dimensional Adjustable Posture-Corrective Orthotic to a Multi-Modal Program for the Treatment of Nonspecific Neck Pain"

_jcm, 2022, doi:10.3390/jcm11237028_

Round 1

Reviewer 1 Report

This study investigated the feasibility and adjunctive efficacy of care program adding adjunctive treatment by a 3D-adjustable posture-corrective orthotic (PCO) to the conventional care program on chronic nonspecific neck pain. This study presented feasibility of the protocol and positive effect of PCO on chronic nonspecific neck pain.

Major comments

1.    2.2. Procedures, line 101: Was assessment for eligibility to include and exclusion conducted prior to obtain the informed consent as described in Figure 1? Or the informed consent was obtained first as mentioned in line 101?

2.    2.2. Procedures, line 105-106; It is described that “. Following the baseline assessment, another research assistant randomized participants to either a study or a control group”, but the randomization described to be conducted prior to the pretreatment assessment in Figure 3. Or does “Pretreatment assessment” refer to Assessment for eligibility in Figure 3?

3.    I'm not sure if my comment is appropriate since there are no units for the changes in Tx and Tz in Table 5 (probably cm). It is concerned that the mean absolute differences within examiners' measurements for lateral and forward translation measurements seem relatively large to changes of these translations in Table 5.

4.    2.5. Interventions, line 255-258: Why walking on a motorized treadmill is necessary for 3D PCO (mirror image therapy)? It would be better to have an explanation as to why.

5.    Table 1 could be improved to make it easier to read and understand.

6.    Figure 4 does not what it says until looking at Figure 5. It would be necessary to devise a diagram that makes it easier to understand what is illustrated by looking at this diagram alone.

7.    Table 3 is shown after Table 4.

8.    In Table 4, for “Integrity and suitability of intervention to participants and Feasibility of study time requirement and study facilities for participants”, participants said the treatment period was too long and hope to reduce treatment period of intervention and treatment sessions as issues of this protocol. Considering the purpose of this study, discussion on this issue would be necessary and provide specific solutions in future study in the discussion.

9.    There does not appear to be any indication such as asterisks to show significant differences in the change within groups (comparing baseline to post-treatment) in Table 5.

10.  It is better to incorporate the results of Tables 6 and 7 into Table 5.

11.  In Table 5, MDs in Study G are minus for ROM flex & extension, ROM lateral flexion, and ROM rotation, are these correct?

12.  In Table 5, MDs of NDI (0-50) in Study G and Control G show positive values, even though the mean value of Post intervention decreased from Baseline. Are these expressions correct?

13.  In Table 6, on the other hand, MDs of NDI (0-50) and NPRS (0-10) are negative even though Study group change score are larger than the Control group change score. It would be better to unify the indication of the increase or decrease of the value.

14.  In Table 6, ROMs in the Study group change score and Control group change score are all negative, does this mean that ROMs decreased after treatment?

15.  It would be better to look at correlations between pre- and post-treatment changes in NDI and Neck Pain and changes in other indicators.

16.  The units of the numbers in the results (Tx, Tz, Rx, Ry, Rz, ROM) in Tables 5, 6, and 7 are not shown.

Minor comments

1.  2.1. Study design, line 90: The different in “different investigators” is unclear what it says.

2.  2.3.1. Inclusion criteria, line 116: It seems unnecessary to use NP as abbreviation for neck pain and in the following text.

3.  2.3.1. Inclusion criteria, line 119: What does GPS stand for here because of the first appearance, although it appeared in line 196. It would be better to have an explanation of what test it is.

4.  2.3.2. and 2.3.3, line 121-138: It is unknown what the Tz, Ry, Rz, Rx, and Tx mean because these are the first appearance. If we read the latter part of the explanation in the section 2.4.4, though, we will understand them.

5.  2.3.3. Posture rotations displacements included the following, line 133: mm is use in line 133 but cm used in line 121 and 122. Thorough the text, figures and tables, it would be better to unify the units as mentioned to use centimeters (or mm) in line 226.

6.  2.4. Examination procedures, line 165: What are special tests?

7.  2.4.1. Primary Outcomes, line 172: What is difference between “study outcome measures” and “primary outcome measures”?

8.  2.4.1. Primary Outcomes, line 172-174: The following sentence seems unnecessary in this section. “We recruited 24 participants with 12 randomly assigned 173 per each of the two groups (33).”

9.  2.4.3. B. Marker placement, line 209: Does “two anterior/posterior views” means one anterior and one posterior or two anterior and two posterior views?

10.  Figure 2. The order of Rx, Ry, Rz, Tx, and Tz in Figure 2 is better to be in order appears in Sections 2.4.5 and 2.4.6 or do it vice versa.

11.  Figure 3: What is “TTT” in the “Study group (n=12) received Conventional TTT T + 3D PCO to make mirror image therapy”. There is a space between “re” and “assessment” in the drop out box.

12.  In Table 2, It would be better to have an explanation of abbreviations use in this figure.

13.  Table 3 and Table 5:  X± SD is better to be mean ± SD in the table, not in the legend. There are and aren’t spaces before and after “±”, and the same for n (%). It is better to unify the decimal places. 1.655 ± 0.5, 1.675 ± 0.6 and 24.2 ± 0.97 would be less misread than 1.655±.5, 1.675±.6 and 24.2±.97. The same for Table 5.

14.  It would be better to align the number of decimal places that appear in the text and figures. For example, in Table 5, ES for Tz Study G is 1.2 but it is described as 1.194 in line 423 in Discussion, although ES for Tx Study G is 1.87 in Table 5 and it is described as 1.87 in line 423.

Reviewer 2 Report

I appreciate the opportunity to review this study. The authors have conducted a study aiming to investigate the feasibility and effect of a multimodal program for the management of chronic nonspecific neck pain CNSNP with the addition of a 3D-adjustable posture-corrective orthotic (PCO), with a focus on patient recruitment and retention.

The manuscript is well-written and detailed; it is apparent great time and effort went into the prepared manuscript. I have some comments about the manuscript which I will list below.

-       Why do the authors consider this a pilot study?

-       I commend the authors for following the CONSORT guidelines;

-       Please, what is the reliability of the postural measurements?

-       Do the authors consider using a control group?

-       Why have you registered the study as a double-blind design (ID: NCT03331120)? In addition, any changes from the registered version should be informed in the manuscript.

-       I commend the authors on the figures, it is much clearer to see the measurements. 
